# Efficacy for Whitlockite for Augmenting Spinal Fusion

**DOI:** 10.3390/ijms222312875

**Published:** 2021-11-28

**Authors:** Su Yeon Kwon, Jung Hee Shim, Yu Ha Kim, Chang Su Lim, Seong Bae An, Inbo Han

**Affiliations:** 1Department of Neurosurgery, CHA University School of Medicine, CHA Bundang Medical Center, Seongnam-si 13496, Gyeonggi-do, Korea; syunkwon@naver.com (S.Y.K.); cslim8112@gmail.com (C.S.L.); 2R&D Center, OSFIRM Co., Ltd., Seongnam-si 13620, Gyeonggi-do, Korea; jhshim@osfirm.co.kr (J.H.S.); yhkim@osfirm.co.kr (Y.H.K.)

**Keywords:** whitlockite, hydroxyapatite, beta-tricalcium phosphate, calcium phosphate ceramic, bone substitute

## Abstract

Whitlockite (WH) is the second most abundant inorganic component of human bone, accounting for approximately 25% of bone tissue. This study investigated the role of WH in bone remodeling and formation in a mouse spinal fusion model. Specifically, morphology and composition analysis, tests of porosity and surface area, thermogravimetric analysis, an ion-release test, and a cell viability test were conducted to analyze the properties of bone substitutes. The MagOss group received WH, Group A received 100% beta-tricalcium phosphate (β-TCP), Group B received 100% hydroxyapatite (HAp), Group C received 30% HAp/70% β-TCP, and Group D received 60% HAp/40% β-TCP (n = 10 each). All mice were sacrificed 6 weeks after implantation, and micro-CT, hematoxylin and eosin (HE) staining, and Masson trichome (MT) staining and immunohistochemistry were performed. The MagOss group showed more homogeneous and smaller grains, and nanopores (<500 nm) were found in only the MagOss group. On micro-CT, the MagOss group showed larger fusion mass and better graft incorporation into the decorticate mouse spine than other groups. In the in vivo experiment with HE staining, the MagOss group showed the highest new bone area (mean: decortication group, 9.50%; A, 15.08%; B, 15.70%; C, 14.76%; D, 14.70%; MagOss, 22.69%; *p* < 0.0001). In MT staining, the MagOss group demonstrated the highest new bone area (mean: decortication group, 15.62%; A, 21.41%; B, 22.86%; C, 23.07%; D, 22.47%; MagOss, 26.29%; *p* < 0.0001). In an immunohistochemical analysis for osteocalcin, osteopontin, and CD31, the MagOss group showed a higher positive area than other groups. WH showed comparable bone conductivity to HAp and β-TCP and increased new bone formation. WH is likely to be used as an improved bone substitute with better bone conductivity than HAp and β-TCP.

## 1. Introduction

Spinal fusion surgery, which combines adjacent vertebrae, is one of the most effective surgical treatments for many types of spinal disorders, including traumatic spine fractures, metastatic spine tumors, spinal deformities, as well as degenerative spinal diseases such as spinal stenosis, herniated intervertebral discs, spondylolisthesis, and spinal instability [1,2,3]. In spinal fusion surgery, bone graft materials are used between adjacent bones to facilitate fusion. Autologous bone graft is the gold-standard material for bone fusion, because it has characteristics necessary for new bone formation, such as osteoconductivity, osteoinductivity, and osteogenicity [4,5,6,7]. However, autologous bone grafts have several complications related to donor-site morbidity, including infection, pain, hematoma, and sensory deficits [8,9,10]. Therefore, many bone graft materials, such as allografts, ceramics, demineralized bone matrix, and growth factor-based substitutes, have been developed to replace autologous bone in terms of structure and function [11,12,13]. Among these bone substitutes, calcium phosphate-based ceramics such as hydroxyapatite (HAp) and beta-tricalcium phosphate (β-TCP), which are biocompatible, biodegradable, and osteoconductive, provide structural support and a chemical environment for new bone formation [12,14,15]. In particular, these ceramics with porous structures have a wide contact surface and provide an environment for angiogenesis where new blood vessels easily grow, improving the fusion rate.

Naturally formed HAp (Ca_10_(PO_4_)_6_(OH)_2_) is the major bone mineral in the human body, constituting 60% of native bone [16]. HAp has a hexagonal crystal structure and maintains bone strength. The structure of HAp is similar to that of the mineral phase of bone and it shows excellent biocompatibility [17,18]. In addition, HAp is the most stable of the calcium phosphate minerals with low solubility in physiological environments [19,20,21]. HAp has similar initial mechanical strength to that of cancellous bone and weak tensile and shear forces. HAp is considered as the best bone alloplast, but its use is limited by its low solubility and rapid decomposition in acidic environments in the body [22].

Along with HAp, β-TCP (Ca_3_(PO_4_)_2_) is one of the two most studied calcium phosphate-based materials. Like HAp, TCP has a highly interconnected, porous structure that can directly assist in fiber vessel penetration and bone replacement [23]. β-TCP is less stable than HAp, but it decomposes faster and is more soluble. It also has a high level of reabsorption, which leads to its replacement by new bone formation. Due to this characteristic of β-TCP, its use for bone regeneration has been actively studied, and β-TCP has been widely used as a bone cement and bone substitute. Unlike HAp, however, β-TCP is not a calcium phosphate-based ceramic that exists in nature; instead, it can only be made through synthesis. HAp has often been mixed with β-TCP to produce biphasic calcium phosphate. When a combination of HAp and β-TCP is used, the different resorption rates of these materials provide orderly bone remodeling over time [15,16].

Whitlockite (WH) is the second most common inorganic component in human bone, comprising about 25% of the human bone based on the amount of Mg^2+^ [24]. Although it accounts for a large proportion of human bone, it has not been used as a bone substitute clinically, primarily due to difficulties in the artificial synthesis of WH and its in vivo detection. Although WH has been relatively underestimated in bone regeneration studies, it has recently been shown to have tremendous potential [24,25]. In acidic conditions (pH < 4.2), WH is known to be more stable than HAp. In previous studies, WH and HAp showed different stability profiles depending on pH conditions [26]. In other words, bone is stable at a wide range of pH levels. Recent studies have discovered that pure WH nanoparticles can be made in acidic aqueous systems with abundant Mg^2+^ ions and low-temperature conditions [22,25,27]. Some studies evaluated the bioactive properties of WH nanoparticles through in vitro and in vivo tests and compared the results with those of HAp and β-TCP [28,29]. WH has also been found to induce higher expression of osteogenic genes than HAp and β-TCP [25,30]. In addition, several studies of in vivo bone regeneration have shown that WH promoted bone growth and osteogenic activity to a greater extent than HAp and β-TCP [28,31,32].

This study investigated the role of WH in bone remodeling and formation in spinal fusion in mice through an interdisciplinary approach to evaluate the effect of this new bone substitute material.

## 2. Results

### 2.1. Morphology and Composition Analysis

The scanning electron microscopy (SEM) observations confirmed the results for the morphology of granules and macropores at lower magnification (Figure 1). Porous granules with homogeneous size and macropores of more than 200 μm were observed in all bone graft products. The higher resolving power of SEM revealed additional details of the nanostructure as well as the microstructure. The MagOss group showed more homogeneous and smaller grains than the other alloplastic products. The grain size of the MagOss group was approximately 300 nm, but the grain size of Groups A and B ranged from 500 nm to 2 μm and that of Groups C and D was more than 1 μm. Micropores of 10~200 μm were observed in all alloplastic products, but nanopores (<500 nm) were only found in the MagOss group because of the different size of the grains.

The crystalline structure of the MagOss group showed the WH pattern, the crystalline structure of Group A showed the β-TCP pattern, and the crystalline structure of Group B showed the HAp pattern. A secondary phase was not found. The peak for the MagOss group displayed a pattern that was more horizontally spread out. The pattern for Groups A and B had a sharper and narrower width than that of the MagOss group.

The X-ray diffractometer (XRD) pattern of the MagOss group showed the WH pattern (Figure 2). The XRD pattern of Group C showed about 30% HAp and about 70% β-TCP, while the XRD pattern of Group D showed about 40% β-TCP and approximately 60% HAp. A secondary phase was not found. The pattern of the MagOss group was more horizontally spread out. The pattern for Groups C and D was sharper and narrower than that of the MagOss group.

### 2.2. Porous Structure and Surface Area

The porosity of the bone graft products was more than 60% (Figure 3). The porosity of the MagOss group was 69.8%, the porosity of Group C was 66.4%, and that of Group D was 82.9%. The pore size distribution as measured by mercury intrusion porosity was found to be similar between Groups C and D, but those materials showed different patterns than those of the MagOss group. The MagOss group was not significantly different from other alloplastic products in terms of macropores and micropores larger than 10 μm. However, nanopores (<500 nm) were only found in the MagOss group.

The MagOss group exhibited high values of surface area, and the Brunauer–Emmett–Teller device (BET) specific surface area was relatively low in Groups C and D. It is possible to observe that the MagOss group exhibited the smallest grain size, which corresponded to a high surface area (5.62 m^2^/g), while Groups C and D exhibited large grains and a low surface area (0.28 and 0.04 m^2^/g).

### 2.3. Thermogravimetry and Differential Thermal Analysis

The total weight loss of the bone graft products, as measured by Thermogravimetry and differential thermal analysis (TG/DTA) to 1300 °C, was found to be less than 5% (Figure 4). The total weight loss of the MagOss group was 0.02%, that of Group C was 0.56%, and that of Group D was 2.83%. It was confirmed that there was no degradation or phase translation to 1300 °C.

### 2.4. Ion Release

Measurements of the number of ions released from the bone graft products for 21 days confirmed that ions were most actively released from the MagOss group (Figure 5).

There was a clear difference in the number of Mg^2+^ ions released between the MagOss and other bone graft products. The MagOss group exhibited continuous release (48.51 ppm for 21 days), whereas Groups C and D showed little release of Mg^2+^ ions (0.27 ppm and 0.50 ppm, respectively). The MagOss group also released P ions more rapidly than Groups C and D (MagOss: 57.36 ppm, Group C: 33.20 ppm, Group D: 9.14 ppm).

In contrast, Ca ion release was higher in Groups C and the D (21.92 and 48.17 ppm, respectively) than in the MagOss group (6.88 ppm). This seems to be because Ca ions and Mg^2+^ ions are released together in the MagOss group, unlike in Groups C and D, where only one type of cation exists. Overall, the total cation release was highest in the MagOss group over the 21 days.

### 2.5. The Effects of WH Bone Grafts on Cell Viability and Proliferation

To evaluate the biocompatibility of the WH bone graft (MagOss), hASCs were cultured in eluted medium derived from each bone graft for 24 h, and cell viability was evaluated using a CCK-8 assay after 24 and 48 h of incubation. As shown in Figure 6, most hASCs showed normal morphology and size. In hASCs, over 90% cell viability (Figure 6) was observed in all bone grafts for 24 or 48 h, and the highest cell viability was observed in the MagOss grafts.

### 2.6. Micro-CT Analysis: In Vivo Experiment

Six weeks after implantation, the incorporation of the bone graft into the recipient lamina was estimated by micro-computed tomography (CT) (Figure 7). The MagOss group showed a larger fusion mass and better graft incorporation into the lamina than other groups. Groups A, B, C, and D showed lower fusion mass and little incorporation into the recipient lamina. The MagOss group demonstrated enhanced bone regeneration and bone bridges at the inter-transverse and articular process areas and the scaffold-implanted sites.

### 2.7. Histological Analysis: In Vivo Experiment

Serial cross-sections (4 μm) were stained with HE and MT to reveal cellular reactions indicative of bone formation at 8 weeks after implantation.

In the quantitative analysis of the new bone area to total bone area ratio using HE staining in the in vivo experimental groups (Figure 8), the MagOss group and Groups A, B, C, and D showed significant differences in the new bone area compared to the decortication group (mean ± SD: decortication group, 9.50% ± 1.34%; A, 15.08% ± 2.34%; B, 15.70% ± 1.19%; C, 14.76% ± 1.50%; D, 14.70% ± 0.95%; MagOss, 22.69% ± 2.67%; *p* < 0.0001). Among these alloplastic product groups, the MagOss group showed the highest new bone area.

In the quantitative analysis of the new bone area to total bone area ratio using MT staining in the in vivo experimental groups (Figure 9), the MagOss group and Groups A, B, C, and D showed significant differences in the new bone area compared to the decortication group (mean ± SD: decortication group, 15.62% ± 2.18%; A, 21.41% ± 3.41%; B, 22.86% ± 3.57%; C, 23.07% ± 2.44%; D, 22.47% ± 1.41%; MagOss, 26.29% ± 3.33%; *p* < 0.0001).

Overall, in the histological analysis including HE and MT staining in the in vivo experiment, the MagOss group showed the highest new bone area to total bone area.

### 2.8. Immunohistochemical Analysis: In Vivo Experiment

Immunohistochemical staining for osteocalcin, osteopontin, and CD31 was performed to evaluate the new bone formation and angiogenesis in the MagOss group and Groups C and D. The average immunoreactivity of osteocalcin showed statistically significant differences between the groups (Appendix A). The MagOss group revealed the highest osteocalcin-positive area (mean ± SD: MagOss, 6.65% ± 1.48%; C, 1.19% ± 1.27%; D, 0.73% ± 0.33%; *p* < 0.0001).

Osteopontin immunohistochemical staining (Appendix A) showed significantly higher expression of osteopontin in the MagOss group compared to Groups C and D (mean ± SD: MagOss, 9.13% ± 2.06%; C, 3.02% ± 1.41%; D, 2.17% ± 0.73%; *p* < 0.0001).

In the quantitative analysis of the CD31-positive area using immunofluorescence in vivo experimental groups (Appendix A), the MagOss group showed a significant difference compared to Groups C and D (mean ± SD: MagOss, 9.06% ± 0.99%; C, 3.84% ± 0.66%; D, 3.39% ± 0.80%; *p* < 0.0001).

## 3. Discussion

This study was designed to prove the potential of WH as a bone graft substitute with higher osteoconductivity, which is a key characteristic of bone substitute materials, than other bone graft materials. It is well-known that the porosity of bone substitute is essential to new bone formation because porosity enables migration and proliferation of osteoblasts and mesenchymal cells, as well as vascularization [33,34]. In this study, the analysis of morphology and composition showed that the WH grains were more homogeneous and smaller than those of other materials, and the WH micropores were smaller than those of other materials, including HAp, β-TCP, and combinations of HAp and β-TCP. In the BET specific surface area analysis of this study, the small grains and pores of WH corresponded to a high surface area. A higher surface area per unit volume of bone graft substitute improved the contact area available for the adhesion and growth of cells, and host tissue ingrowth, including vasculature [34].

WH is a calcium phosphate-based ceramic that has a chemical formula of Ca_9_Mg(HPO_4_)(PO_4_)_6_ [35]. Different Ca/P ratios and solubility can cause different amounts of free Ca^2+^, Mg^2+^, and P^+^ ions to be released (WH = 1.43, HA = 1.67, β-TCP = 1.5). In the ion-release analysis of this study, WH continuously released higher levels of Mg^2+^ and P^+^ ions than HAp and β-TCP under physiologically relevant conditions. Mg is the fourth most abundant cation in the human body and is naturally found in bone, where approximately half of the total Mg is stored [36,37]. In addition, Mg is essential to human metabolism, and a lack of Mg, in general, is associated with decreased osteoblastic and osteoclastic activity, osteopenia, and vitamin D reduction [38]. In several prior studies, the release of Mg^2+^ ions from WH has been shown to be an advantage over HAp in terms of efficient osteogenic differentiation. Calcium phosphate compounds containing Mg^2+^ have been reported to promote cell adhesion, proliferation, and differentiation, thus improving biocompatibility and bone formation [39,40]. In vitro experiments conducted by Jang et al. using human-derived osteoblasts showed significantly higher growth with WH than with HAp and TCP, and the expression of osteogenesis genes was similar or slightly higher in WH [24]. In addition to the intrinsic properties of WH, they demonstrated that the better biocompatibility of WH might result from many factors, such as nanostructure, mechanical hardness, and roughness. The osteoinductivity of bone substitutes can be evaluated by examining whether new bone forms after implantation at a heterotopic site. Calcium and phosphorus decomposed from ceramic surfaces are known to improve the gene expression of osteoinductive substances such as bone morphogenetic proteins, thereby facilitating bone marrow stem cells to differentiate into osteoprogenitor cells, and contributing to the formation of new bone [41]. In the quantitative study using micro-CT and histologic analyses, WH showed higher new bone formation than other bone substitutes. In the WH group, new bone was found with higher connectivity and thickness than in other groups. This result indicates that WH has characteristics that promote bone formation. Further, the immunohistochemical analysis results showed higher levels of osteocalcin, osteopontin, and CD31, which may suggest that WH facilitated bone formation and vascularization. Similar to our study, in a rabbit ilium defect model [31], after 6 weeks, the WH group had compatible percent bone volume and significantly thicker bone with a more directional form than the HAp and β-TCP groups. Likewise, in a rat calvarial defect model [28], porous WH/chitosan promoted bone regeneration significantly better than a HAp/chitosan composite scaffold. Therefore, these findings imply that WH has higher osteoconductivity than other materials, which is a key component of bone substitute materials.

To the best of our knowledge, there are few previous experiments of spinal fusion models for the effectiveness of WH. In the current study, in vitro and in vivo analyses were performed simultaneously to investigate the effect on spinal fusion in a mouse spine model along with the physical and chemical properties of WH. In particular, in animal experiments, WH was proven to be effective in new bone formation and fusion through micro-CT analysis, histological analysis using HE and MT staining, and immunohistochemical analysis using osteocalcin, osteopontin, and CD31.

There are several limitations to our study. We did not perform the three-point bending test in the animal experiment to qualify the mechanical strength and elastic deformation. We also used immunohistochemical analysis with osteocalcin, osteopontin, and CD31 only in WH and biphasic calcium phosphate groups, and not in HAp and β-TCP groups. This study was conducted on small animals. To further verify the effectiveness of WH in spinal fusion, additional experiments with spine models in medium- to large-sized animals are necessary. In addition, WH may be clinically applied through further clinical trials in the near future.

## 4. Materials and Methods

### 4.1. Synthetic Bone Grafts

The following groups were created with different bone grafts: MagOss (0.6–1.0 mm particle size, 100% WH bone graft, OSFIRM Co., Ltd., Seongnam, Gyeonggi-do, Korea), Group A (1.0–2.0 mm, 100% β-TCP), Group B (1.0–2.0 mm, 100% HAp), Group C (0.6–1.0 mm, 30% HAp, 70% β-TCP), and Group D (0.6–1.0 mm, 60% HAp, 40% β-TCP). All grafts were commercially available bone graft substitutes. Excelos^®^ (100% β-TCP), Bongros^®^ (100% HAp), and Boncel-Os^®^ (30% HAp/70% β-TCP) were obtained from CGbio Company Limited (Seongnam, Korea). Osteon^TM^ (60% HAp/40% β-TCP) was purchased from Genoss Company Limited (Suwon, Korea).

The MagOss group was prepared by sintering polyurethane sponges infiltrated with the WH slurry containing vehicle, binder, and dispersant. Specifically, the vehicle and binder were mixed with a dispersant to form the first slurry. The first slurry prepared was deagglomerated by stirring and subsequently deaired until there was no further release of air bubbles. The WH powder was added to the first slurry to form the second slurry with the following ratio: WH powder:first slurry = 1:1.2. The second slurry was mixed using a planetary mixer and a 3-roll-mill until completely mixed. The polyurethane foam was cut into the desired shape and size and immersed in the second slurry for coating. The second coating was applied by dipping after first sintering at 750 °C for 2 h to eliminate the matrix. After drying, the samples were sintered again at 750 °C for 2 h to increase the compressive strength and crystallinity of porous bodies. Finally, the sintered bodies were crushed to 0.6~1.0 mm and were sterilized by gamma irradiation.

### 4.2. Characterization of Bone Grafts

#### 4.2.1. Scanning Electron Microscopy

The morphology of the bone graft products was investigated by SEM, using a Philips XL-30-FEG device at an accelerating voltage of 15 kV.

#### 4.2.2. X-ray Diffractometer

The phase composition of the bone graft products was determined using an XRD (Smart Lab, Rigaku, Germany). The XRD patterns were recorded at room temperature using Cu Kα1 radiation with the following measurement conditions: tube voltage of 40 kV, tube current of 40 mA, step size of 0.02, and a scan speed of 3°/min over an angular range from 20° to 80°.

#### 4.2.3. Porosity

The porosity and pore size diameter were obtained using a mercury porosimeter (MicroActive AutoPore V 9600, Micromeritics Instruments Inc., Norcross, GA, USA) in the pressure range of 0.20 to 33,000 psia.

#### 4.2.4. Assessment of the Surface Area of Bone Graft Products

The surface area of bone graft products was assessed using a BET (TristarII 3020, Micromeritics Instruments Inc., Norcross, GA, USA) with N_2_ adsorption for a pore size distribution of 2 to 300 nm.

#### 4.2.5. Thermogravimetry and Differential Thermal Analysis

TG-DTA of the bone graft products were carried out with a simultaneous thermal analyzer–mass spectrometer (STA 409 PC, NETZSCH, Selb, Bavaria, Germany) at a heating rate of 10 °C/min from room temperature to 1300 °C.

#### 4.2.6. Ion Release of the Bone Graft

The ion release of the bone graft products was assessed by measuring the concentrations of Ca^2+^, Mg^2+^, and P^+^ ions released from the specimens versus immersion time via inductively coupled plasma atomic emission spectrometry (ICP-AES, OPTIMA8300, PerkinElmer, Waltham, MA, USA). A 1.0 g sample of the bone graft products was immersed in 5 mL of deionized water, with a specimen chosen for each immersion time (30 min and 1, 2, 3, 7, 14, and 21 days). At each time point, an aliquot of 0.5 mL was removed and replaced by a fresh solution.

### 4.3. Cell Viability of Bone Graft

#### CCK8 Assay

Human adipose-derived stem cells (hASCs) were purchased from Lonza (PT-5006, Lot.0000605220). hASCs were maintained in Dulbecco’s Modified Eagle’s Medium (DMEM, Gibco^®,^ Thermo Fisher Scientific, lnc., Walthan, MA, USA) with 10% Fetal Bovine Serum (FBS, Gibco^®,^ Thermo Fisher Scientific, lnc., Walthan, MA, USA) and 100 units/mL of penicillin–streptomycin (Gibco^®^) in a humidified 37 °C incubator with 5% CO_2_. hASCs (passage 5) were seeded at a density of 1 × 10^5^ cells/well with culture media onto a 12-well plate. After 24 h of culture, the culture medium was replaced with an extract medium which was eluted for 24 h by soaking each bone graft. The control groups were exposed only to standard culture medium. After 24 or 48 h of incubation, cell viability was evalutated using a CCK-8 assay (Dojindo Molecular Technologies, Inc., Rockville, MD, USA). CCK reagent was added per well and incubated at 37 °C for 3 h. The absorbance values were measured at 450 nm using a microplate reader (Epoch™ Microplate Spectrophotometer, BioTek Instruments, Inc., Winooski, VT, USA). All experiments were performed in triplicate wells for each group.

### 4.4. In Vivo Experiment with Bone Grafts

#### 4.4.1. Animal Experiment

The animal experiments were performed in accordance with the approval of the Institutional Animal Care and Use Committee of CHA University (IACUC200058). Twelve-week-old female C57BL/6 mice weighing 20 g were purchased from Koatech Company Limited (Pyeongtaek, Korea) and raised under conditions of 55–65% humidity and a controlled temperature of 24 ± 3 °C with a light/dark cycle of 12 h. The mice had free access to food and tap water ad libitum. The experiment was conducted in duplicate and the animals were randomly divided into experimental groups. In the experiment, the MagOss group received WH (n = 10), Group A received 100% β-TCP (n = 10), Group B received 100% HAp (n = 10), Group C received 30% HAp/70% β-TCP (n = 10), and Group D received 60% HAp/40% β-TCP (n = 10).

Mice were placed under anesthesia with a mixture of zoletil (50 mg/kg; Virbac Laboratories, Carros, France), Rompun (10 mg/kg; Bayer, Seoul, Korea), and saline solution injected intraperitoneally. The surgical site covered with hair and skin was shaved with a blade and disinfected with povidone-iodine and 70% ethanol. After a 15 mm incision of the skin and fascia was performed over the L4–L6 spinous processes, posterolateral lumbar spine fusion surgery was performed with a 1 mm pneumatic diamond burr from L4 to L6 to decorticate the articular processes until punctate bleeding was induced. The surgical procedures were performed by bilaterally revealing the articular processes of L4–L6. In each group, bone graft substitutes were implanted over the bilateral decorticated articular processes at each site. The fascia and skin were sutured with 4–0 black silk. Mice were cared for until full recovery on a heating pad following the surgical procedures. All mice were sacrificed by carbon dioxide inhalation 6 weeks after implantation, and their spines were resected for analysis.

#### 4.4.2. Micro-CT Analysis

We acquired micro- CT images with Skyscan1173 (Bruker-microCT, Kontich, Belgium) at a resolution of 13.85 μm (achieved using 130 kV and 60 μA). A 1.0 mm-thick aluminum filter was used and the total exposure time was 500 ms. The acquired data were reconstructed with Nrecon software (Ver. 1.7.4.6, Bruker micro-CT, Kontich, Belgium). The grayscale threshold values for the graft particles and newly formed bone were standardized, and the corresponding grayscale values were within the ranges of 126–225 and 70–125, respectively. The augmented bone was measured in terms of the total region of interest volume, the volume of residual graft material, and newly formed bone.

#### 4.4.3. Histology and Immunohistochemistry

After micro-CT scanning, specimens were decalcified using 10% EDTA buffered with Tris-EDTA solution (pH 7.2–7.4, to 700 mL of PBS, add 100 g EDTA and adjust pH as needed to 7.2–7.4 by cautiously adding drops of 10 N NaOH) for 2 weeks. Before being embedded into a paraffin block, the plastic tube was removed. Those blocks were sectioned at a thickness of 4 μm and then stained with hematoxylin/eosin (HE) and Masson trichrome (MT). Histological images were acquired with a digital slide scanner (Paranoramic 250 Flash III, 3d-Histech, Budapest, Hungary). The acquired 3D images were measured using an image analysis program (Image Pro Plus, Media Cybernetics, Inc., Rockville, MD, USA). For immunohistochemistry, the sections were incubated overnight in the primary antibody at 4 °C. The next day, the sections were stained with Alexa Fluor^®^ 488 goat anti-rabbit IgG (H + L) (Invitrogen, Waltham, MA, USA, A10034) for 1 h at room temperature. Primary antibodies used in staining were rabbit-polyclonal to osteocalcin (mouse-specific, 1:100, Abcam, Cambridge, UK, ab93876), rabbit-polyclonal to osteopontin (mouse/human-specific, 1:100, Abcam, Cambridge, UK, ab8448), and rabbit-polyclonal to CD31 (human/mouse/pig-specific, 1:50, Abcam, Cambridge, UK, ab28364), which were used to demonstrate new bone formation and angiogenesis.

### 4.5. Statistical Analysis

The results were expressed as mean ± standard deviation and statistically examined using one-way analysis of variance followed by Tukey’s post hoc analysis using GraphPad Prism software (version 8; GraphPad Inc., La Jolla, CA, USA). The results were considered statistically significant at *p* < 0.05.

## 5. Conclusions

Based on quantitative and qualitative analyses, WH can induce bone regeneration to its original state through phase transformation. Overall, these findings indicate that WH minerals can be applied to bone healing because they can stimulate bone regeneration. WH also showed bone conductivity comparable with that of HAp and β-TCP and increased bone movement on the surface. As a result, WH is likely to be used as a better bone substitute with better bone conductivity than HAp and β-TCP.

## Figures and Tables

**Figure 1 ijms-22-12875-f001:**
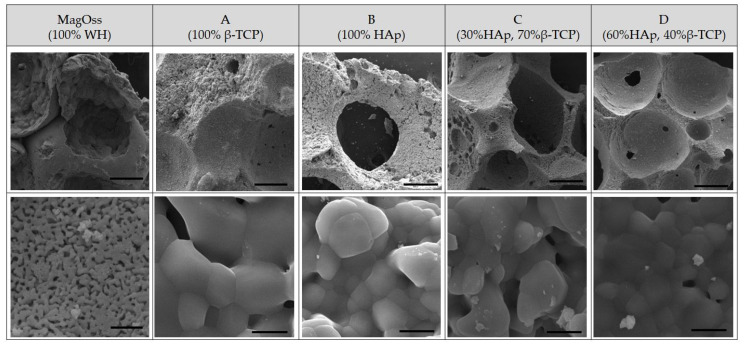
SEM images of different bone grafts. Macropores of more than 200 μm and Micropores of 10~200 μm were observed in all bone graft products. However, nanopores (<500 nm) were only found in the MagOss group. The grain size of the MagOss group was approximately 300 nm, but the grain size of Groups (**A**,**B**) ranged from 500 nm to 2 μm and that of Groups (**C**,**D**) was more than 1 μm. Scale bar = 200 μm, 2 μm.

**Figure 2 ijms-22-12875-f002:**
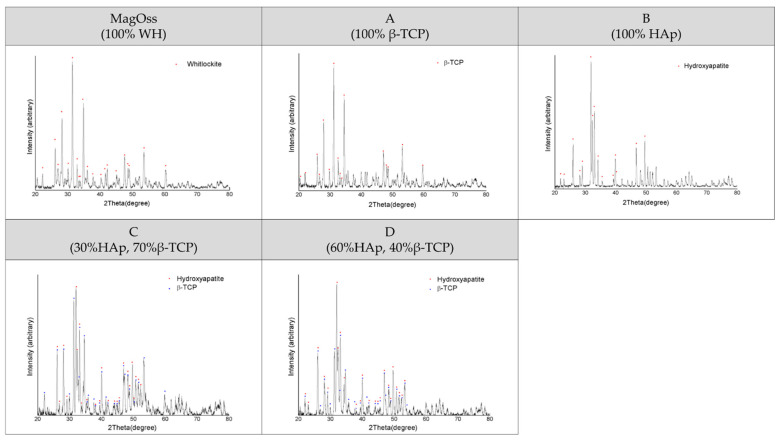
XRD patterns of different bone grafts. These observations were obtained by X-ray diffraction (XRD) analysis. A secondary phase was not found in any of the bone graft products. The XRD pattern of the MagOss group was broader than that of the others. Group (**A**) showed the β-TCP pattern, Group (**B**) showed the HAp pattern, Group (**C**) showed about 30% HAp and about 70% β-TCP, while Group (**D**) showed about 40% β-TCP and approximately 60% HAp. A secondary phase was not found. The pattern for Groups (**A**–**D**) had a sharper and narrower width than MagOss group.

**Figure 3 ijms-22-12875-f003:**
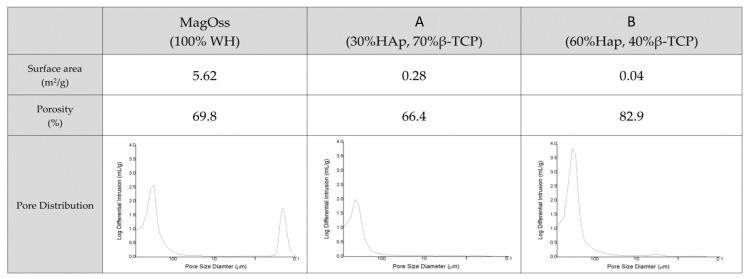
Surface area and pore size distributions of different bone grafts. The porosity of bone graft products was more than 60%, but nanopores (<500 nm) were only found in the MagOss group. The MagOss group exhibited the smallest grain size, which corresponds to a high surface area. The surface area was relatively low in Groups (**A**,**B**).

**Figure 4 ijms-22-12875-f004:**
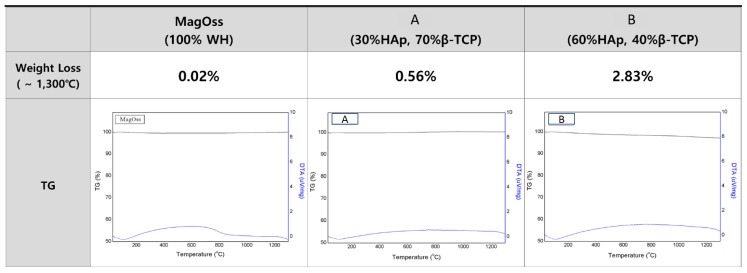
TG/DTA patterns of different bone grafts. The total weight loss of bone graft products as measured by thermogravimetric analysis to 1300 °C was found to be less than 5%. The total weight loss of the MagOss group was 0.02%, that of Group (**A**) was 0.56%, and that of Group (**B**) was 2.83%. It was confirmed that there was no degradation or phase translation to 1300 °C.

**Figure 5 ijms-22-12875-f005:**
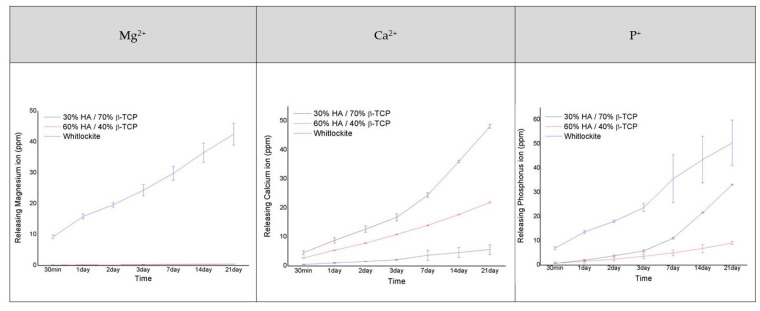
Analysis of the amount of Mg^2+^, Ca^2+^, and P^+^ ions released (as ions related to osteogenic differentiation).

**Figure 6 ijms-22-12875-f006:**
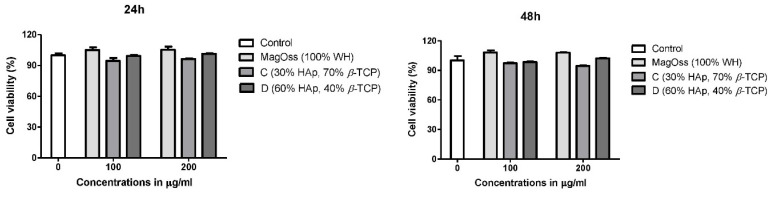
Effect of bone grafts on cell viability of human adipose-derived stem cells.

**Figure 7 ijms-22-12875-f007:**
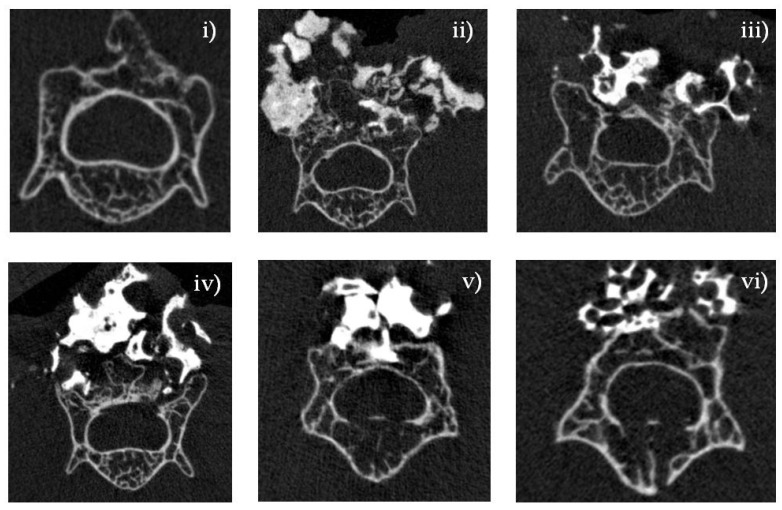
Micro-CT analysis of bone regeneration in a spinal fusion model. Three-dimensional reconstructed images and cross-sectional images of micro-CT at 6 weeks. Bone histomorphometry of 3D bone formation architecture. MagOss group showed better graft incorporation into the recipient lamina. (**i**) Decortication, (**ii**) MagOss (100% WH), (**iii**) A (100% β-TCP), (**iv**) B (100% HAp), (**v**) C (30% HAp + 70% β-TCP), (**vi**) D (60% HAp + 40% β-TCP).

**Figure 8 ijms-22-12875-f008:**
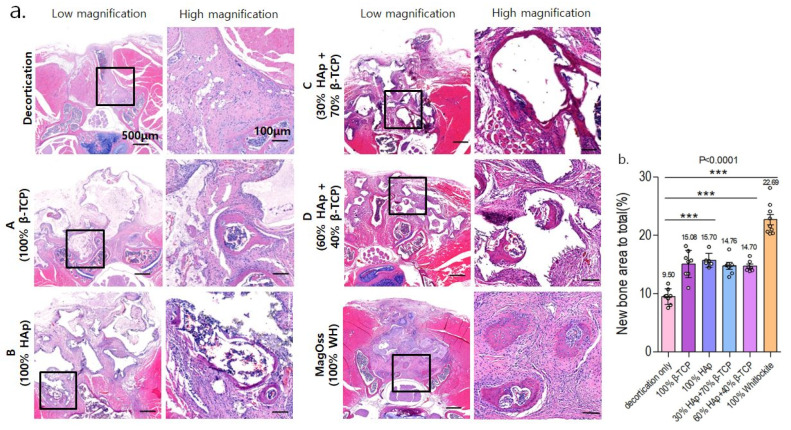
Histological analysis of bone regeneration in a spinal fusion model using hematoxylin and eosin staining. (**a**) Images of cross-sections from each group (scale bar = 500 μm, 100 μm). (**b**) Quantitative measurements of new bone area to total bone area (*** *p* < 0.0001). Data were expressed as the mean ± SE (decortication, A, B, MagOss: n = 9/C, D: n = 7).

**Figure 9 ijms-22-12875-f009:**
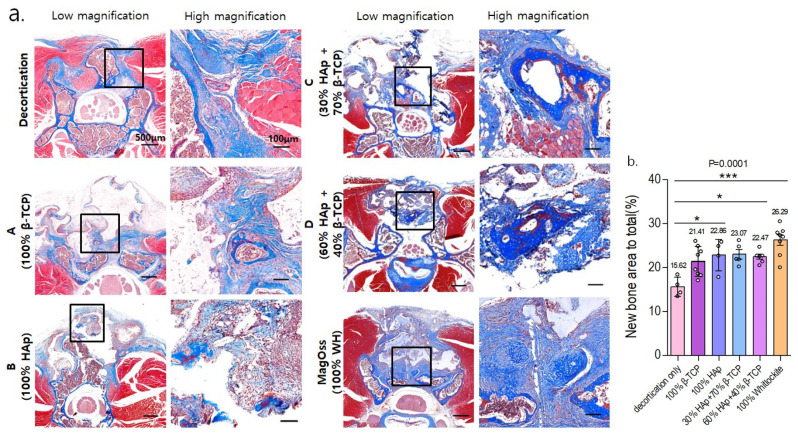
Histological analysis of bone regeneration in a spinal fusion model using Masson trichrome staining. (**a**) Images of cross-sections from each group (scale bar = 500 μm, 100 μm). (**b**) Quantitative measurements of new bone area to total bone area (* *p* < 0.05, *** *p* < 0.0001). Data were expressed as the mean ± SE (decortication, A, B, MagOss: n = 8/C, D: n = 5).

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
