# Peer review of "Efficacy for Whitlockite for Augmenting Spinal Fusion"

_ijms, 2021, doi:10.3390/ijms222312875_

Round 1

Reviewer 1 Report

General Comments

The authors are using whitlockite to generate scaffolds for bone regeneration to treat mouse spinal defects.  Their description of the production and characterization of scaffolding material is adequate.  However, the cellular component of the study has serious problems.

Specific Comments

Materials

The authors used ASCs, but failed to provide information regarding these cells.  What species were they derived from?  Were they purchased or obtained from an internal source?  How where they cultured and maintained prior to the use?  This information is essential for the interpretation of the results.

The ASCs were cultured in eluted medium derived from bone grafts for 24 hours.  What does this mean and what is the rationale?

Figure 6 is useless and does not provide clear images of cells.

Mice were care for, what does this mean?

Histology

Tissue decalcified with EDTA, what was the percentage of EDTA and what was the buffer?

Immunohistochemistry for 10 min.  This is a very short incubation period.

Need to describe antibodies: polyclonal/monoclonal? Mouse Specific? Etc.  Were secondary antibodies used? Controls?

Reviewer 2 Report

Abstract: well written.

Introduction: Too long. Should be reduced.

M&M: Line 164: Write number of approval from ethical committee.

Results: well written and presented. No further improvements.

Discussion: Write about the strengths and limitation of your paper. Please discuss further improvements. here are some inspiration for discussion (doi: 10.2147/IJN.S205880 and doi: 10.17305/bjbms.2019.3854).

Conclusion. Well written.

Round 2

Reviewer 1 Report

The text has been satisfactorily modified.  However, the photographs in Figure 6 are not sufficient quality for publication.

Author Response

Thank you for pointing it out. As your advice, we tried to get better quality photos, but unfortunately failed. So we decided to remove the image of cell morphology, and leave only the statistical graph. Also check all English spellings. Thank you.